# Beyond Trauma-Induced Coagulopathy: Detection of Auto-Heparinization as a Marker of Endotheliopathy Using Rotational Thromboelastometry

**DOI:** 10.3390/jcm13144219

**Published:** 2024-07-19

**Authors:** Alexandru Emil Băetu, Liliana Mirea, Cristian Cobilinschi, Ioana Cristina Grințescu, Ioana Marina Grințescu

**Affiliations:** 1Department of Anesthesiology and Intensive Care II, Carol Davila University of Medicine and Pharmacy, 050474 Bucharest, Romania; alexandru.baetu@gmail.com (A.E.B.); ioana.grintescu@rospen.ro (I.M.G.); 2Department of Anesthesiology and Intensive Care, Grigore Alexandrescu Clinical Emergency Hospital for Children, 011743 Bucharest, Romania; 3Department of Anesthesiology and Intensive Care, Clinical Emergency Hospital Bucharest, 014461 Bucharest, Romania; 4Department of Anesthesiology and Intensive Care, Zetta Clinic, 020311 Bucharest, Romania; ioana_grintescu@yahoo.com

**Keywords:** multiple trauma, trauma-induced coagulopathy, auto-heparinization, thromboelastometry, critical care

## Abstract

**Background/Objectives**: The complexity of trauma-induced coagulopathy (TIC) is a result of the unique interactions between the patient, trauma, and resuscitation-related causes. The main objective of trauma resuscitation is to create the optimal milieu for both the development of immediate reparatory mechanisms and the prevention of further secondary injuries. Endotheliopathy represents one of the hallmarks of trauma-induced coagulopathy, and comprises endothelial dysfunction, abnormal coagulation, and inflammation, all of which arise after severe trauma and hemorrhagic shock. **Methods**: We retrospectively and descriptively evaluated 217 patients admitted to the Bucharest Clinical Emergency Hospital who met the Berlin criteria for the diagnosis of multiple trauma. Patients with high suspicion of auto-heparinization were identified according to the dynamic clinical and para-clinical evolution and subsequently tested using rotational thromboelastometry (ROTEM). The ratio between the clot formation time (CT) was used, obtained on the two channels of interest (INTEM/HEPTEM). **Results**: Among the 217 patients with a mean age of 43.43 ± 15.45 years and a mean injury severity score (ISS) of 36.98 ± 1.875, 42 patients had a reasonable clinical and para-clinical suspicion of auto-heparinization, which was later confirmed by the INTEM/HEPTEM clotting time ratio in 28 cases (12.9% from the entire study population). A multiple linear regression analysis highlighted that serum lactate (estimated 0.02, *p* = 0.0098) and noradrenaline requirement (estimated 0.03, *p* = 0.0053) influenced the CT (INTEM/HEPTEM) ratio. **Conclusions**: There is a subset of multiple trauma patients in which the CT (INTEM/HEPTEM) ratio was influenced only by serum lactate levels and patients’ need for vasopressor use, reinforcing the relationship between shock, hypoperfusion, and clotting derangements. This emphasizes the unique response that each patient has to trauma.

## 1. Introduction

Endotheliopathy of trauma (EOT) comprises endothelial dysfunction and anomalous coagulation and inflammation, which arise after severe trauma and hemorrhagic shock and represent one of the hallmarks of TIC [1]. A unifying model of shock-induced endotheliopathy (SHINE) describes the maladaptive changes induced by the activation of the sympathetic nervous system, independent of the etiology of the shock. It involves vasoconstriction, the activation of endothelial cells, glycocalyx degradation, the destruction of junctions between endothelial cells, an increase in vascular permeability, the formation of a procoagulant vascular microenvironment, a further reduction in the tissue oxygen supply as a result of edema and micro-thrombosis, and the activation of leukocytes, all of which lead to end-organ dysfunction and increased mortality [2,3]. Furthermore, the aggressive resuscitation means used in treating shock states could potentially lead to a further worsening of endothelial activation and organ dysfunction, ultimately resulting in resuscitation-associated endotheliopathy (RAsE) [4].

Markers of glycocalyx degradation and subsequent endothelial dysfunction have been analyzed for their intrinsic role as TIC mediators through the different effects they exert on platelet function, thrombin generation, or fibrinolysis [5]. 

The glycocalyx is a gel-like structure that lines endothelial cells and is an essential modulator of endothelial function, vascular permeability, inflammation, hemostasis, fibrinolysis, signaling, and cellular interactions [6,7]. Its structure includes glycolipids, glycoproteins, and proteoglycans. Proteoglycans (e.g., syndecan 1–4, glypican 1–6) are structures grafted on the ectodomain of transmembrane proteins that have branched polysaccharide chains composed of 1–4 linked repeating units of β-D-glucuronic acid, N-acetyl-glucosamine, and disaccharides with different degrees of sulphation, called glycosaminoglycans (heparan sulfate, chondroitin sulfate, hyaluronic acid) [7,8,9,10].

A release into the circulation of fragments that are structurally similar to heparin (i.e., heparan sulfate) leads to the auto-heparinization or endogenous heparinization phenomena, which have been documented by viscoelastic hemostatic assets (VHA) such as thromboelastography (TEG) and ROTEM, with conflicting results [11,12,13].

It is worth mentioning that mast cells contribute to both local and systemic inflammatory responses induced by hemorrhagic shock [14,15]. They store and synthesize many substances, including interleukin 6, tumor necrosis factor, histamine, serotonin, substance P, tryptase, chymase, other specific proteases, and heparin. There is a complex relationship between mast cell degradation and coagulation. Studies on anaphylaxis have shown that the severity of anaphylaxis is correlated with the increase in heparin and bradykinin formation, contact system activation, and fibrinolysis caused by tryptase-induced plasmin activation. However, the impact of endogenous heparinization produced by mast cell degradation in multiple trauma is yet to be established [16].

Ostrowski et al. investigated endogenous heparinization by TEG, using kaolin, kaolin–heparinase and plasma levels of sympathetic nervous system activation, damage to endothelial cells, and glycocalyx, inflammation, coagulation, and fibrinolysis markers as activators, respectively, as determined by classical methods. In their study, 5% of patients with severe post-traumatic injuries showed signs of auto-heparinization, as assessed by TEG and increased levels of syndecan-1. This subgroup of patients had higher transfusion requirements, increased prothrombin time values (expressed as an international normalized ratio (INR)), hyperfibrinolysis, and protein C deficiency [13]. 

Zipperle et al. analyzed auto-heparinization using ROTEM and used ellagic acid on the INTEM channel and ellagic acid and heparinase on the HEPTEM channel as activators. The results showed that the injury severity and the presence of hemorrhagic shock were not associated with an increase in CT on the INTEM channel in comparison with HEPTEM, and the increased levels of heparan sulfate did not lead to significant differences in CT on the two channels. Heparin administration resulted in a significant prolongation of CT/INTEM, as expected. Despite the structural similarities between heparin and heparan sulfate, the latter did not show a detectable anticoagulant effect in ROTEM analysis [11].

A possible explanation for the differences between the results of the VHA may lie in the particularities of the methods and reactants. In both assays, whole blood is added to a cup in which a pin is suspended. In TEG, the cup filled with the blood sample is rotated (in a limited arc ±4°45′ every 5 s) and engages a pin transduction system as clot formation occurs, while in ROTEM, the cup is not mobile, and the pin transduction system oscillates (±4°45′ every 6 s). The end effect is the formation of a clot that will affect the rotation of the assembly, thus allowing for an analysis of the kinetics of clot formation, strength, and lysis [12]. Both VHAs use negatively charged surfaces that act as classical intrinsic pathway activators and a combination of the same activators and heparinase, which neutralizes heparin or heparin-like substances, thus revealing the clotting profile in the absence of the heparin anticoagulant effect. Heparinases catalyze the depolymerization reactions of heparin or heparan sulfate. They are classified into three major types: heparinase I cleaves highly sulfated heparin or heparan sulfate chains, and heparinase III cleaves fewer sulfated chains. Heparinase II cleaves high and low sulfation domains. Combining the three heparinases can produce the near-complete depolymerization of heparin or heparan sulfate polysaccharide chains to disaccharides [17]. In opposition to TEG, all tissue factor-activated channels in ROTEM contain polybrene, a heparin inhibitor able to neutralize small amounts of heparin. It remains to be studied whether this influences the ability of the two VHAs to detect auto-heparinization. According to the manufacturers, both TEG and ROTEM assays use heparinase I from flavobacteria. It is currently unknown whether the type of heparin lyases impact HEPTEM in trauma patients [18,19].

We hypothesized that a simultaneous evaluation of heparan sulfate inhibition by heparinase and polybrene could be used to detect auto-heparinization in trauma patients by analyzing the clotting time ratio from INTEM/HEPTEM.

## 2. Materials and Methods

We conducted a retrospective, descriptive study that consisted of two parts. First, we analyzed all patients admitted in the last five years to the Emergency Clinical Hospital Bucharest with a diagnosis of multiple trauma. Second, we analyzed the demographic and para-clinical data of the patients who had a reasonably high suspicion for auto-heparinization 24 h after the traumatic event.

Patients were diagnosed with multiple trauma based on the Berlin criteria. The maintenance of an adequate cardiac output was guided by minimally invasive hemodynamic monitoring, in association with repeated cardiac ultrasound evaluations. All patients had a central venous catheter and an arterial catheter placed either in the Emergency Department or immediately after admission to the ICU. Continuous invasive blood pressure was measured in all patients by arterial cannulation, the transmission of the pressure waveform of the arterial pulse to a pressure transducer, and conversion into an electrical signal that was processed, amplified, and converted into a visual display [20,21]. Pulse pressure variation (PPV) was evaluated in patients with sinus rhythm, mechanically ventilated with fixed tidal volumes, and without thoracic trauma or an open chest. Fluid resuscitation was performed using crystalloid solutions, and vasopressor and inotropic support followed adequate fluid resuscitation whenever required. We guided the resuscitation phase based on the ROSE concept: mean arterial pressure (MAP) > 65 mm Hg, cardiac index (CI) > 2.5 L min^−1^m^−2^, PPV < 12%, left ventricular end-diastolic area index (LVEDAI) > 8 cm m^−2^, bearing in mind the deleterious effects of hypervolemia and hemodilution on RaSE, and so we decided on a low threshold for initiating vasopressor and inotropic support, as per the latest European guidelines on the management of major bleeding and coagulopathy following trauma. Noradrenaline was added to fluids to maintain target arterial pressure whenever the use of a restricted volume replacement strategy did not achieve the target blood pressure, and dobutamine was added in the presence of myocardial dysfunction, either clinically suspected whenever there was a poor response to fluid expansion and norepinephrine, or as determined by transthoracic echocardiography CI < 2.5 L min^−1^m^−2^ [21,22,23,24].

The acid-base disorders were aggressively treated, and normothermia was promptly reached and maintained. A key point in managing the patients was reaching a hemostatic balance. Initially, patients received liberal blood transfusions, i.e., red blood cells (RBC)–fresh frozen plasma (FFP) 1:1, but after obtaining the results of the first thromboelastometric assay, they were switched to a goal-directed algorithm.

Patients with reasonably high clinical suspicion for auto-heparinization were defined as having a decrease in blood hemoglobin of more than 0.5 g/dL at 24 h after the traumatic event in the absence of hypervolemia ((ruled out with ultrasonography: left ventricular filling pressures (E/e’) ultrasonographic evaluation, diameter of inferior vena cava > 2 cm, hepatic vein Doppler with reversed systolic wave, portal vein Doppler with pulsatility ≥ 50%, renal parenchymal vein Doppler with monophasic diastolic wave pattern [25,26]) and hyperfibrinolysis (ruled out using thromboelastometric assay: maximum lysis or ML/EXTEM < 15%). In our analysis, we considered auto-heparinization to be present in patients with a CT (INTEM/HEPTEM) ratio greater than 1.25, as previously defined by Gorlinger et al. in orthotopic liver transplants or severe trauma patients, and further discussed below [27,28]. 

Patients under chronic anticoagulant medication treatment and those with chronic liver disease or diastolic disfunction were subsequently excluded from the study. A seric calcium level outside the normal range and initial resuscitation following a massive transfusion protocol also led to exclusion. Patients who underwent emergency surgical procedures, such as damage control surgery, were also excluded.

The viscoelastic device used in this study was a rotational thromboelastometer-ROTEM Sigma (ROTEM^®^; TEM International, Munich, Germany).

The statistical analysis was performed with the GraphPad 10 Prism program. Data normality was tested using the D’Agostino–Pearson test. The Wilcoxon test for paired data was used to compare the effectiveness of initial resuscitation, and the Mann–Whitney test for unpaired data was used to compare the auto-heparinized group with the non-auto-heparinized group. A one-sample *t*-test analysis was performed for parametric data, which compared the mean of the CT (INTEM/HEPTEM) ratio with the hypothetical value of 1 (the null hypothesis that the coagulation time would be equal on the two channels). The multiple linear regression used the CT ratio as the dependent variable and clinical and para-clinical data obtained from patient admission as the independent variables.

The study was conducted in accordance with the Declaration of Helsinki and approved by the Clinical Emergency Hospital of Bucharest Ethics Committee (protocol code 11097/03.12.2019) for the data collection, analysis, and publishing of the results.

## 3. Results

This study included patients (*n* = 217, 129 male and 88 female) with a mean age of 43.43 years old and a standard deviation (95% CI) of 15.45 years. Even if the inclusion criteria were based on the Berlin definition, assessing the ISS was mandatory. The mean ISS value was 36.98 ± 1.875. The physical mechanisms that led to multiple trauma were mostly car crashes (58.25%, *n* = 127), followed by falls from height (19.40%, *n* = 42), work accidents (16.35%, *n* = 35), and sports accidents (6%, *n* = 13).

Advanced Trauma Life Support was initiated in the pre-hospital setting and subsequently continued and completed in the Emergency Clinical Hospital Bucharest according to the European guidelines on the management of major bleeding and coagulopathy following trauma.

Patient management efficacy was analyzed using the comparative analysis of the mean values of interest data using the Wilcoxon test for paired data. Follow-up was performed after 6 h by reevaluating coagulation parameters with classical laboratory tests and arterial blood gas for acid-base disorders. 

At six hours after the initiation of acid-base and volume resuscitation, an analysis of arterial blood gas and acid-base analysis, a hemogram, and a coagulogram were performed. A comparison of the median pH showed a correction at 6 h (7.30 vs. 7.36, *p* < 0.0001), depicted in Figure 1 and Table 1. The base excess (BE) (mmol/L) at 6 h was also lower than recorded at admission (−10 vs. −2.8, *p* < 0.0001), depicted in Figure 2 and Table 1. The lactate level (mmol/L) followed the acid-base trend, with the follow-up level having a lower median value compared to the starting level (2.4 vs. 1.8, *p* < 0.0001), depicted in Figure 3 and Table 1. Serum bicarbonate (mmol/L) increased as a result of the specific treatment instituted, registering a higher level at 6 h (19 vs. 21.6, *p* < 0.0001), depicted in Figure 4 and Table 1. The analysis of classic parameters of coagulation emphasized the decrease in the INR value (1.64 vs. 1.1 *p* < 0.0001), depicted in Figure 5, Table 1, and the partial thromboplastin time (PTT) value (seconds) (37 vs. 34, *p* < 0.0001), depicted in Figure 6 and Table 1, after the administration of blood products and derivatives, according to the European guidelines on management of major bleeding and coagulopathy following trauma. The hemoglobin level (g/dL) was also corrected, with the median level detected at 6 h being statistically significantly higher than the initial level (9.2 vs. 9.5, *p* < 0.0001), depicted in Figure 7 and Table 1.

Cardiocirculatory instability was detected by minimally invasive hemodynamic monitoring and mean artery pressure was used as a cut-off. A value of less than 65 mmHg led to vasopressor administration, specifically noradrenaline, in 118 patients (55.75%). After that, the patients were evaluated with heart ultrasound. Cardiac output measurements determined the use of inotropic drugs, specifically dobutamine, in 21 cases (9.6%).

Advanced life support was continued, utilizing specific intensive care maneuvers. The paraclinic monitoring in the next 24 h showed a decrease in hemoglobin levels without simultaneous detection of fibrinolysis by VHA. 

Since our center could not measure syndecan 1–4 levels to confirm auto-heparinization, we used ROTEM, more specifically the HEPTEM assay, to estimate the ratio between INTEM CT and HEPTEM CT (Figure 8).

After verifying the normal distribution with the D’Agostino–Pearson test (K2 value = 4.138, *p* = 0.12), a one-sample t-test was performed, which used the hypothetical comparison value of 1 (the null hypothesis that the ratio of the coagulation time investigated using INTEM and HEPTEM would be equal). The mean value of the CT (INTEM/HEPTEM) ratio for the study group was 1.23. Thus, with a significance level of *p* < 0.0001, the null hypothesis was rejected.

A value above 1.25 of the CT (INTEM/HEPTEM) ratio was considered sufficient to demonstrate auto-heparinization. Of the total 41 patients, 28 (12.9% of the entire study population) had a value of this ratio above 1.25. Even though thromboelastometric analysis must answer the question: “Why is this patient bleeding?” and not the question “Will this patient bleed?”, the results of studies investigating auto-heparinization have been inconsistent. Thus, a multiple linear regression analysis, using the CT (INTEM/HEPTEM) ratio as the dependent variable and the paraclinical data of the patient since admission to the intensive care unit of the Bucharest Clinical Emergency Hospital (BE, serum lactate, serum bicarbonate, pH, hemoglobin, INR, PTT, ISS, use of noradrenaline or dobutamine) as the independent variables, becomes mandatory (Figure 9 and Figure 10).

The normality of residuals was tested with the D’Agostino–Pearson omnibus (k2), which highlighted the normal distribution of the data (*p* = 0.62). Multicollinearity was tested using the variance inflation factor (VIF) for each coefficient. Given that the highest VIF value was 2.06, multicollinearity and the presence of redundant information were excluded. The accuracy of the prediction of the linear regression model can be evaluated on the Y-axis (Figure 10). Considering the fact that the value of residuals is the difference between the observed and predicted values, any positive value means an underestimation of the CT ratio (INTEM/HEPTEM), and any negative value means an overestimation of CT ratio (INTEM/HEPTEM). However, even if it is not ideal, our plot is pretty symmetrically distributed and tended to cluster around the lower single digit of the Y-axis. From the variance analysis of the studied parameters, serum lactate (with an estimate of 0.02, *p* = 0.0098) and the need for noradrenaline to maintain MAP (with an estimate of 0.03, *p* = 0.0053) were the only variables that seemed to influence the CT (INTEM/HEPTEM) ratio.

The description of the multiple trauma patients group considered auto-heparinization (CT INTEM/HEPTEM ratio > 1.25) by assessing the number of RBC and FFP, but also the length of intensive care unit stay (LOIS) and length of hospital stay (LOHS) in comparison with the rest of the patients included in the study (multiple trauma patients without confirmation of auto-heparinization (*n* = 189)). Patients considered auto-heparinized received at least one unit of RBC after the initial resuscitation phase, while patients without proven auto-heparinization did not always receive a blood transfusion during their hospital stay. However, the median value was similar, with both groups needing two units of RBC. The Mann–Whitney test did not reveal a statistically significant difference: *p* = 0.07. In the case of the administration of FFP, even though it was not necessary in all cases of polytraumatized and auto-heparinized patients, the median value detected was two units of FFP compared to only one unit for the rest of the patients. This difference was statistically significant: *p* < 0.0001. LOIS and LOHS were similar. Even though the Mann–Whitney test did not highlight a statistically significant difference in the comparative analysis of the number of intensive care days required, it is still important to note that patients with multiple trauma who had a CT INTEM/HEPTEM ratio > 1.25 had a median value of 14 days, vs. 9 days in the group of patients not considered auto-heparinized (Table 2).

## 4. Discussion

In recent decades, the prognosis of a patient with multiple trauma has improved primarily through understanding the systemic response mechanisms to tissue injury. Thus, survival has been improved by anticipating and avoiding late complications in these patients. In the “pre-hospital” phase or in the first 24 h after the traumatic event, mortality has remained quasi-similar. Bleeding and coagulopathy, often self-perpetuating in a vicious circle, have remained the cause of most deaths [29].

The complexity of coagulopathy is due to the unique interactions between the patient, trauma, and causes associated with resuscitation. The patient presents a series of particularities such as age, comorbidities, background drug treatment, and genetic anomalies that are the basis of a pro or anticoagulant profile pre-existing the trauma. The actual trauma leads to tissue destruction, hemorrhage, and shock, with the consequence of hypoperfusion, which is pathophysiologically translated by systemic endotheliopathy. At this level, mechanisms such as the activation of the sympathetic nervous system, damage to the glycocalyx, auto-heparinization, an inflammatory response, platelet dysfunction, the decreased activity of coagulation factors, and hyperfibrinolysis occur. They accumulate with the reduction in coagulation factors through loss and consumption. Additionally, factors related to resuscitation measures, such as the dilution of coagulation factors, hypothermia, and acidosis, contribute to the aforementioned dysfunctions. Each of these elements constitutes a potential stopping point in the vicious circle of TIC, a direction of diagnostic research, and a possible therapeutic target [19].

For a very long time, the physicians treating these patients had to address the “lethal triad” represented by acidosis–coagulopathy–hypothermia. Understanding the pathophysiology of a patient in the hyperacute phase following multiple trauma gave way to hypocalcemia. To the best of our knowledge, there has been no report of serum calcium as an independent risk factor in multiple trauma. However, hypocalcemia associated with the classic triad leads to increased mortality in the hyperacute phase. According to Roshini Pradeep et al., modern reports on the prognosis of polytraumatized patients began to include calcium, leading to the development of the “lethal diamond” [30].

The main objective of trauma resuscitation is to create the optimal milieu for both the development of immediate reparatory mechanisms and the prevention of further secondary injuries. We address the “trauma diamond of death” using pH-guided fluid resuscitation and mechanical ventilation for normocapnia, thus preventing acidosis, the aggressive forestalling and treatment of hypothermia, the VHA-guided correction of hyperfibrinolysis, platelet dysfunction, fibrinogen depletion and decreased thrombin generation, and the correction of trauma-related and transfusion-related hypocalcemia. We guide the administration of fluids and cardioactive and vasoactive substances using hemodynamic assessment and develop individualized end-organ perfusion goals based on lesional pattern and patient pre-existent conditions.

When reviewing algorithms for ROTEM-guided bleeding management, Gorlinger et al. considered a CT (INTEM/HEPTEM) ratio greater than 1.25 to be diagnostic for auto-heparinization in liver transplant patients. Severe heparin-like effect (HLE), auto-heparinization, or endogenous heparinization (CT INTEM/HEPTEM ≥ 1.25) were associated with increased transfusion requirements, and when they occurred during the anhepatic phase, they were associated with an increased 3-month mortality [27]. Ranucci’s group, when investigating the phenomenon in patients treated with heparin-free extracorporeal membrane oxygenation using TEG, defined HLE as at least one analysis showing a reaction time (R-time) with heparinase at least 30% shorter than the correspondent R-time without heparinase. In their study, patients with HLE had a significantly higher rate of septic complications, and the release of endogenous heparin-like substances from both the glycocalyx and the mast cells was considered to be the result of the systemic inflammatory reaction triggered by contact of the blood with foreign surfaces or by sepsis [31]. 

The comprehensive literature review and original research by Zipperle et al. consistently showed that auto-heparinization could not be demonstrated using ROTEM, but could be with TEG. As the authors stated, the ellagic acid assay might not be sufficiently sensitive to detect heparin-like substances released during the EOT, and they could not exclude that other intrinsically activating reagents besides heparan sulfate might play a role [11].

To our understanding, despite using different VHA methods, the differences between our results and Zipperle’s, as well as the similarities to Ostrowski’s, can be explained by patients’ characteristics, translated into the degree of shock and SHINE, as depicted by the high levels of endogenous catecholamines or the high need for exogenous catecholamines [11,13].

In our study, we were able to identify a subset of polytrauma patients without an active bleeding source, adequately volume and acid-base resuscitated, with residual coagulopathy and without a fibrinolytic phenotype. These patients, recognized by a minor decrease in the level of hemoglobin, arrived at the Bucharest Emergency Clinical Hospital with some major metabolic disorders, and in the vast majority of cases they required vasopressor support, and sometimes also inotropic. 

We are fully aware that the simple ratio between CT obtained on INTEM and HEPTEM does not certify auto-heparinization in the absence of a statistical analysis that also includes the transmembrane structures (structurally similar to heparin) of glycocalyx degradation (e.g., syndecan 1–4). Given the fact that, so far, we are not aware of a trauma center that can measure syndecan levels on a regular basis, the focus must be on the patient and the local possibilities of diagnosis and treatment. However, this study, based on clinical judgment, has an advantage compared to the inconsistent reports on this subject completed prior [11,13]: the volemic, acid-base, and fluid-coagulant resuscitation before thromboelastometry analysis. 

Unfortunately, the treatment modality of this possible endotheliopathy remains unclear. The natural approach would be the administration of FFP, known and demonstrated as having hemostatic, anti-inflammatory, and endothelial permeability-lowering effects [32,33].

On the other hand, assumptions or recommendations for using protamine have also been raised, given the similarity that the transmembrane structures released into circulation following EOT have to heparin [1,32]. The answer to these questions will only be elucidated with further research, but it is imperative to keep clinical judgment at the heart of any treatment: Is this patient bleeding? Do they have an active source? Have dilutional coagulopathy, hyperfibrinolysis, thrombin generation deficiency, or fibrinogen deficiency been ruled out? Have the constituents of the lethal diamond been ruled out? Do we have a reasonable suspicion of auto-heparinization? Is the auto-heparinization self-limiting?

The practical impact auto-heparinization has on a multiple trauma patient remains to be determined. However, this retrospective analysis touched on four important aspects: the number of units of blood and fresh frozen plasma, length of intensive care unit stay, and length of hospital stay. Beyond the pre-hospital phase, the initial resuscitation and the critical first 48 h in intensive care are just the beginning of the “road” of multiple trauma patients. Often, they need reparative surgery (orthopedics, general surgery, plastic surgery, and reconstructive microsurgery). For the most part, after the initial resuscitation phase, the administration of RBC and FFP takes place in the operating rooms during these reparative interventions. Even though the number of units of RBC administered to patients considered auto-heparinized was not higher compared to non-auto-heparinized patients and no spontaneous bleeders could be identified, the auto-heparinized patients required more FFP, perhaps reflecting the higher degree of endothelial damage. While it is beyond the scope of the present study and the study design is not suitable for forming definite conclusions, we can still raise the alarm regarding patients who have a reasonable suspicion of auto-heparinization after initial resuscitation. Patients with a CT (INTEM/HEPTEM) ratio > 1.25 are not spontaneous bleeders, but represent a potentially bleeding group.

Shock and hypoperfusion have been systematically incriminated in the maintenance of endotheliopathy associated with multiple trauma, yet treatment indications are still subject to research. Although the frequency of shock in the polytrauma patient is unknown, it is an independent risk factor for coagulopathy associated with multiple trauma. Thus, maintaining a MAP of at least 65 mmHg is essential, but cardiocirculatory management should be guided by advanced hemodynamic monitoring and dynamic echocardiography. Moreover, in our study, the CT (INTEM/HEPTEM) ratio, used as a control of auto-heparinization, was influenced only by the patient’s need for noradrenaline and the serum lactate level, as witnesses to the presence of the shock.

## 5. Conclusions

In conclusion, the present study highlighted the presence of possible auto-heparinization using the ROTEM viscoelastic test in 28 patients (12.9%) beyond volemic, acid-base resuscitation, and correction of initial coagulopathy. The patient’s need for noradrenaline and serum lactate levels as a proxy for shock represented the only determinants of the CT (INTEM/HEPTEM) ratio.

## Figures and Tables

**Figure 1 jcm-13-04219-f001:**
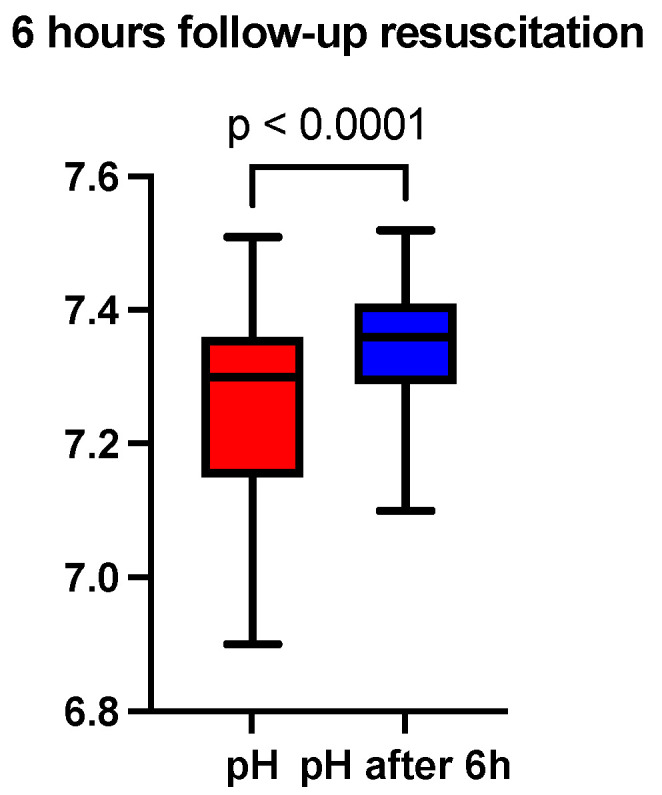
Six-hour follow-up resuscitation—median comparison of pH.

**Figure 2 jcm-13-04219-f002:**
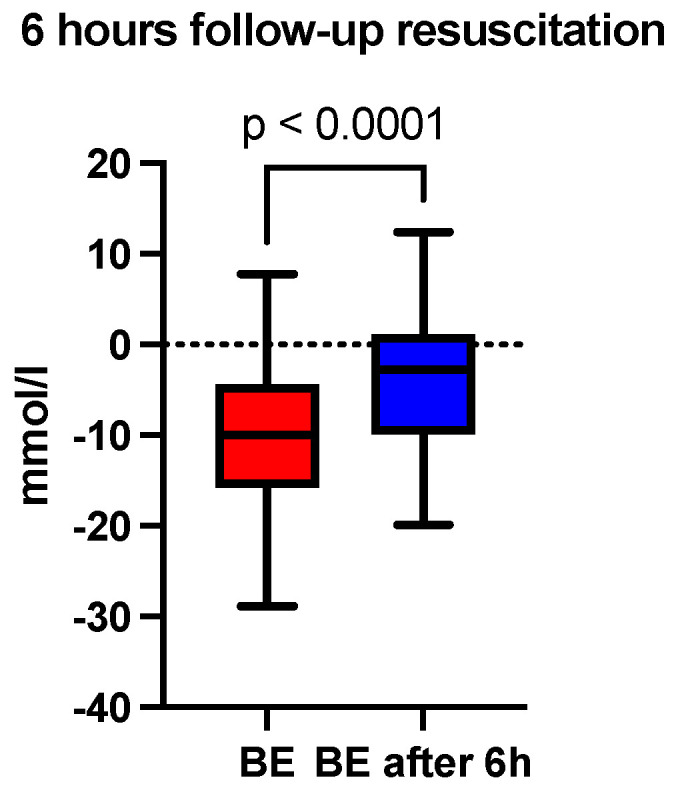
Six-hour follow-up resuscitation—median comparison of the base excess.

**Figure 3 jcm-13-04219-f003:**
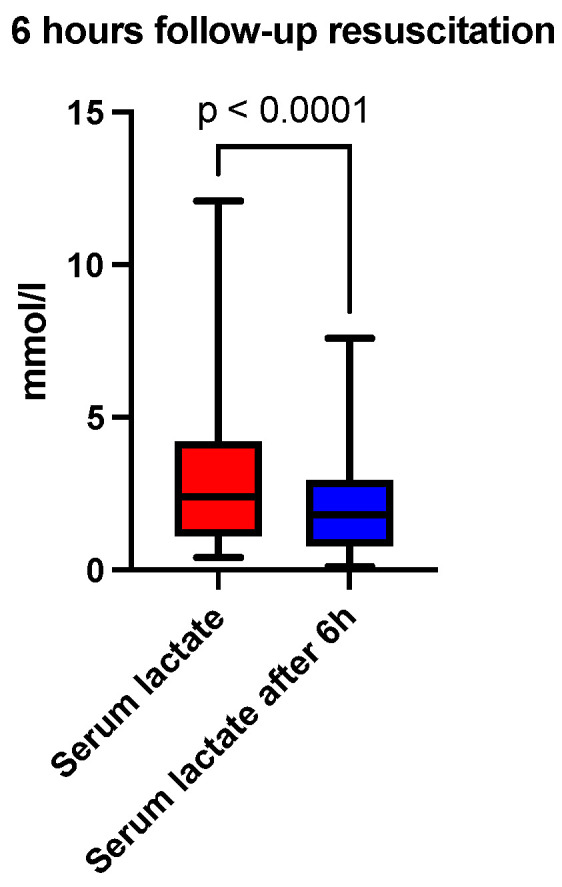
Six-hour follow-up resuscitation—median comparison of serum lactate.

**Figure 4 jcm-13-04219-f004:**
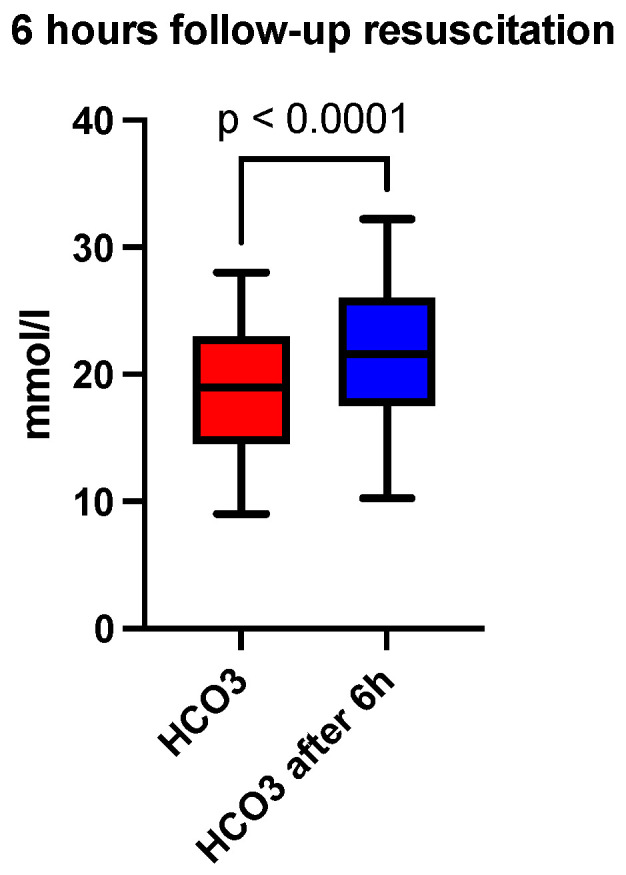
Six-hour follow-up resuscitation—median comparison of serum bicarbonate.

**Figure 5 jcm-13-04219-f005:**
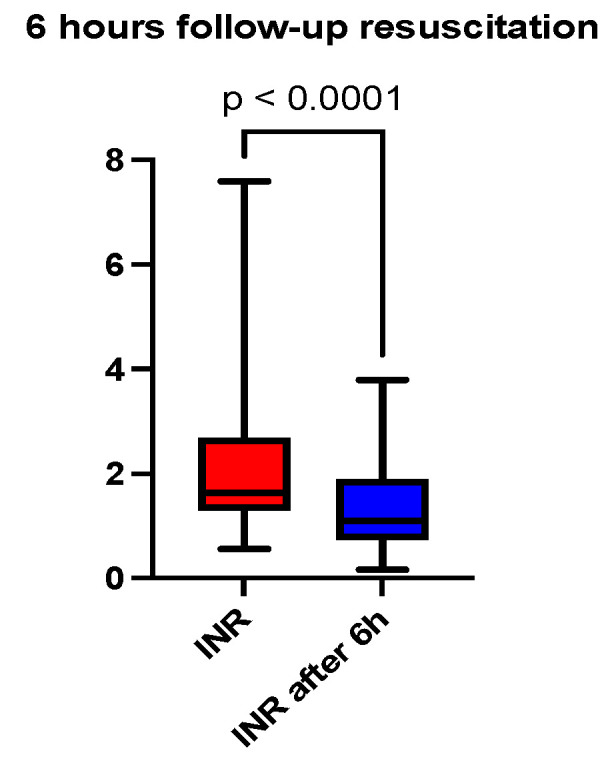
Six-hour follow-up resuscitation—median comparison of INR.

**Figure 6 jcm-13-04219-f006:**
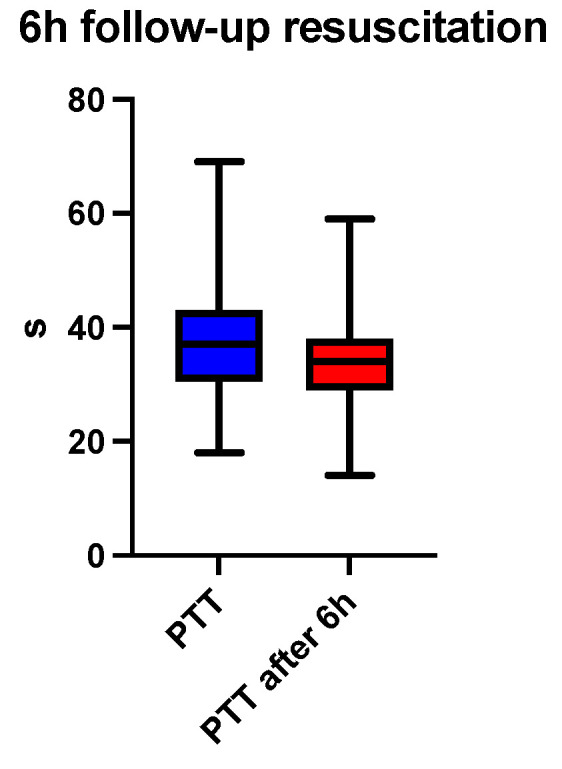
Six-hour follow-up resuscitation—median comparison of PTT.

**Figure 7 jcm-13-04219-f007:**
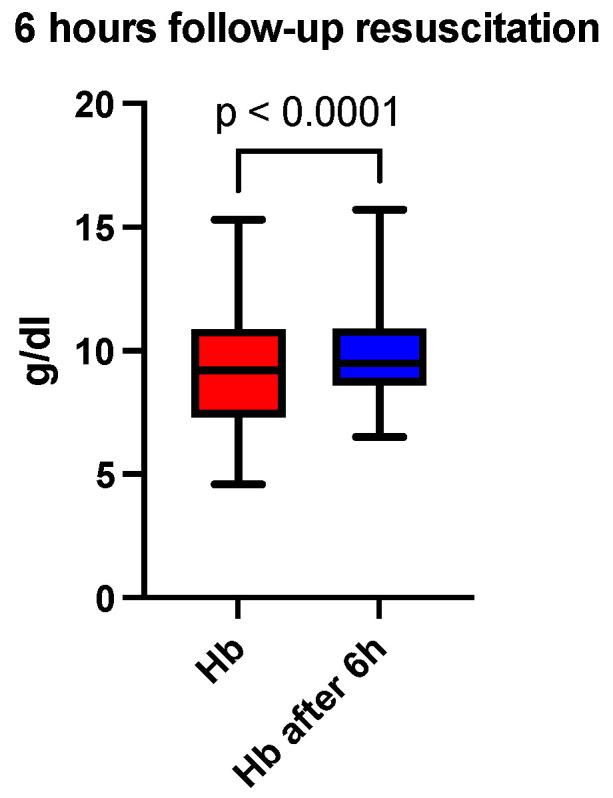
Six-hour follow-up resuscitation—median comparison of the hemoglobin level.

**Figure 8 jcm-13-04219-f008:**
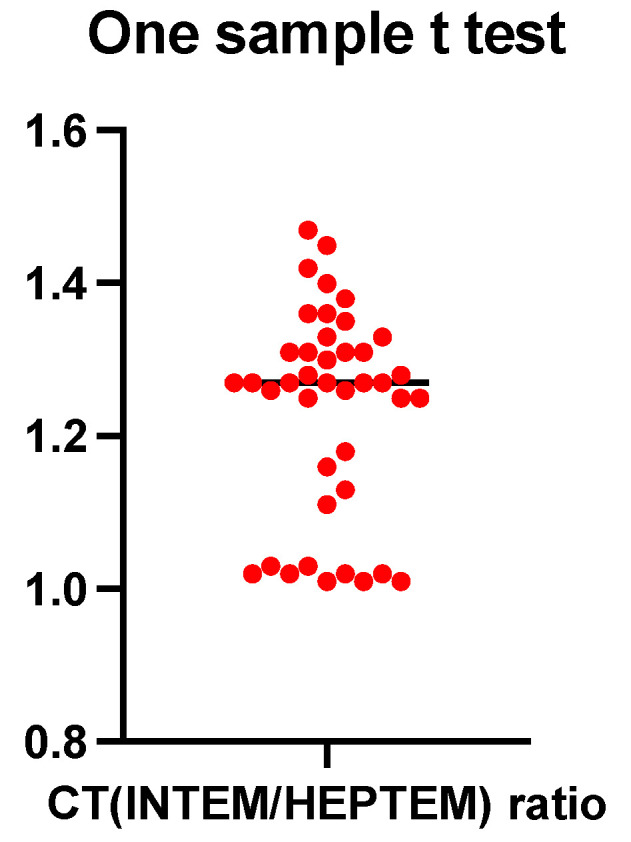
One-sample *t*-test for the CT (INTEM/HEPTEM) ratio.

**Figure 9 jcm-13-04219-f009:**
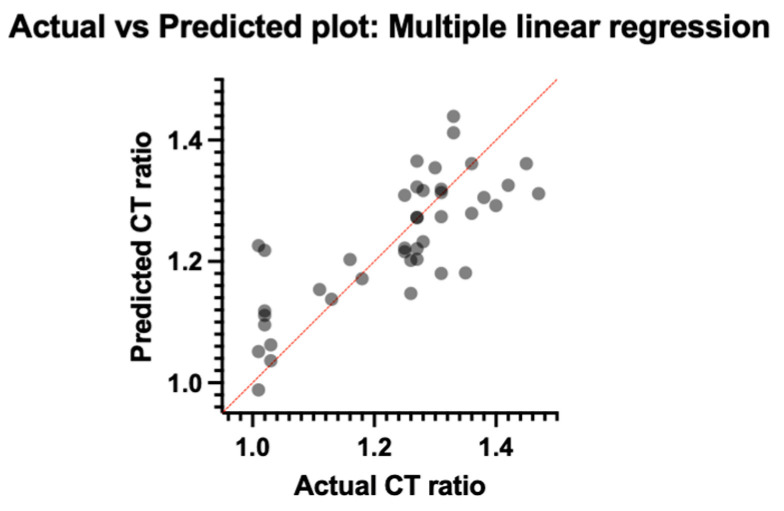
Actual vs. predicted plot CT ratio (INTEM/HEPTEM).

**Figure 10 jcm-13-04219-f010:**
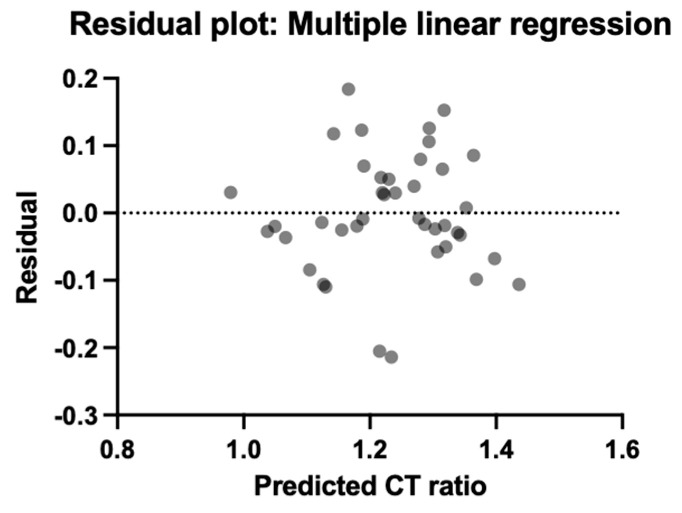
Residual vs. predicted CT ratio (INTEM/HEPTEM).

**Table 1 jcm-13-04219-t001:** Evaluation of the initial resuscitation of multiple trauma patients.

Characteristics	Baseline on Admission—25% Percentile	Baseline on Admission—Median	Baseline on Admission—75% Percentile	6 h Follow-Up—25% Percentile	6 h Follow-Up—Median	6 h Follow-Up—75% Percentile	*p*-Value—Two Tailed
pH	7.15	7.3	7.36	7.29	7.36	7.41	<0.0001
Base excess (mmol/L)	−15.8	−10.00	−4.4	−9.9	−2.8	1.1	<0.0001
Serum lactate (mmol/L)	1.1	2.4	4.2	0.78	1.8	2.9	<0.0001
Serum bicarbonate (mmol/L)	14.5	19	23	17.5	21.6	26	<0.0001
Hemoglobin (g/dL)	7.3	9.2	10.9	8.6	9.5	10.9	<0.0001
INR	1.29	1.64	2.6	0.73	1.1	1.9	<0.0001
PTT(s)	30.5	37	43	29.00	34	38	<0.0001

**Table 2 jcm-13-04219-t002:** Comparative analysis of multiple trauma patients considered auto-heparinized versus multiple trauma patients not considered auto-heparinized.

Characteristics	Minimum Value—Auto-Hep Group	Median Value—Auto-Hep Group	Maximum Value—Auto-Hep Group	Minimum Value—NO-Auto-Hep group	Median Value—NO-Auto-Hep Group	Maximum Value—NO-Auto-Hep Group	*p*-Value—Two Tailed
RBC units	1	2	5	0	2	6	0.07
FFP units	0	2	3	0	1	3	<0.0001
LOIS (days)	3	14	34	6	9	34	0.44
LOHS (days)	3	18	44	6	19	36	0.52

## Data Availability

All presented data are available on demand.

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
