# Peer review of "Beyond Trauma-Induced Coagulopathy: Detection of Auto-Heparinization as a Marker of Endotheliopathy Using Rotational Thromboelastometry"

_jcm, 2024, doi:10.3390/jcm13144219_

Round 1
Reviewer 1 Report
Comments and Suggestions for Authors
Baetu et al. present a manuscript where they investigated the poorly understood trauma induced auto-heparinization phenomenon. There a few remarks this reviewer would like to make:
1. Authors need to provide further insight into those "28 patients with auto-heparinization". How many of them were bleeders, how many of them received transfusion of blood or blood products?
2. It would be also advisable to discuss the mast cell degranulation in contexts of auto-heparinization.
3. Citations need to be corrected throughout the manuscript, and added where necessary (e.g., sentence starting in line 43 requires reference).
Comments on the Quality of English LanguageSome English editing is required for the manuscript (e.g., Paragraph staring at line 70 sounds odd).
Author Response
Thank you for the suggestions and problems raised, the solutions of which can only lead to the improvement of the article. We also corrected the text using MDPI English Editing services.
Attached you will find our answer!

Reviewer 2 Report
Comments and Suggestions for Authors
see attached word document
These are detailed comments

see attached document
Author Response

(The authors gave the same response as above.)

Round 2
Reviewer 2 Report
Comments and Suggestions for Authors
The authors should place the scatter points on the graphics 9 and 10.
See attached basic example which is attached.

Author Response
Dear reviewer,
We have updated in the article the required regression graphs (9 and 10) in JPG format. We hope that there will be no more viewing problems now. The scatter plot is a standard one width of distribution of points proportionate to the number of points at Y value.
We thank you for all the time and effort you put into giving us constructive and helpful comments, which have significantly impreved the quality and clarity of our manuscript!